# COVID-19 Down Under: Australia’s Initial Pandemic Experience

**DOI:** 10.3390/ijerph17238939

**Published:** 2020-12-01

**Authors:** Matthew James Cook, Gabriela Guizzo Dri, Prishanee Logan, Jia Bin Tan, Antoine Flahault

**Affiliations:** 1Global Studies Institute, University of Geneva, 1205 Geneva, Switzerland; Gabriela.Guizzo@etu.unige.ch (G.G.D.); Prishanee.Logan@etu.unige.ch (P.L.); Jia-Bin.Tan@etu.unige.ch (J.B.T.); 2Melbourne School of Population and Global Health, University of Melbourne, Bouverie Street, Carlton, VIC 3053, Australia; 3Institute of Global Health, Faculty of Medicine, University of Geneva, 1205 Geneva, Switzerland; Antoine.Flahault@unige.ch

**Keywords:** Australia, case study, novel coronavirus, COVID-19, acute respiratory disease, epidemiology, non pharmaceutical intervention, country economy, social political disruption, media coverage, mathematical modelling, exit strategies

## Abstract

The following case study aims to provide a broad overview of the initial Australian epidemiological situation of the novel coronavirus disease (COVID-19) pandemic. We provide a case presentation of Australia’s current demographic characteristics and an overview of their health care system. The data we present on Australia’s COVID-19 situation pertain to the initial wave of the pandemic from January through to 20 April 2020. The results of our study indicate the number of reported COVID-19 cases in Australia reduced, and Australia initially managed to successfully flatten the curve—from an initial doubling time of 3.4 days at the end of March 2020 to a doubling time of 112 days as of 20 April 2020. Using SEIR mathematical modelling, we investigate a scenario assuming infections increase once mitigation measures are lifted. In this case, Australia could experience over 15,000 confirmed cases by the end of April 2020. How Australia’s government, health authorities and citizens adjust to preventative measures to reduce the risk of transmission as well as the risk of overburdening Australia’s health care system is crucial. Our study presents the initial non-pharmaceutical intervention measures undertaken by the Australian health authorities in efforts to mitigate the rate of infection, and their observed and predicted outcomes. Finally, we conclude our study by presenting the observed and expected economic, social, and political disruptions Australians may endure as a result of the initial phase of the pandemic.

## 1. Introduction

With increased globalization and an ever-growing population, global health experts had warned it was a matter of time before the next pandemic—a biological threat—“whose speed and severity rivalled those of the 1918 influenza epidemic,” would occur [1,2]. On 29 December 2019, that threat became real. News surrounding the fruition of the novel coronavirus disease, COVID-19, emerged in Wuhan, China.

During this time, Australia was in the midst of dealing with catastrophic fires as well as extreme flooding and a brutal drought. News of this potentially virulent strain did not make Australian headlines until 25 January 2020, when the World Health Organisation (WHO) reported 1320 confirmed cases of COVID-19 globally, four of which were in Australia [3].

It was evident that the spread of the severe acute respiratory syndrome coronavirus 2 (SARS-Cov-2) virus, which causes COVID-19, was and would be exponential. By 31 January 2020, with over 9800 cases of COVID-19 globally, the WHO declared the 6th “Public Health Emergency of International Concern (PHEIC)” [4]. On 11 March 2020 with more than 100,000 global cases of COVID-19 and over 4000 deaths, the WHO declared a global pandemic [5].

The COVID-19 pandemic will be an enduring global challenge. Current data suggest that the virus is approximately 10 times more severe than influenza, with a case fatality rate between 0.25–10%, depending on location, and age (with higher rates for older populations and those with underlying health conditions) [6]. Without governments and citizens taking appropriate measures to suppress and contain the virus, the average infected person is most likely to spread the disease to at least two to three other persons [7]. A number of different approaches have been adopted by different countries to slow the spread of COVID-19, thus relieving the burden on healthcare systems and ultimately saving lives. China, the epicentre of COVID-19, imposed strict quarantine measures and a nationwide lockdown; Singapore did not impose a lockdown but did impose heavy fines on citizens who did not abide by the government’s quarantine measures [8].

As of 20 April 2020, the outbreak of COVID-19 had spread to 213 countries and claimed over 160,000 lives with over 2.4 million confirmed cases globally [7]. Australia had approximately 6600 confirmed cases and had recorded just over 70 deaths [9]. In order to slow the spread of infection, the World Health Organization (WHO) has recommended a combination of measures: rapid identification and immediate isolation of positive cases, and rigorous contact tracing and self-quarantine for those who have been potentially exposed to someone who tested positive for COVID-19. Ultimately, how each country deals with the COVID-19 pandemic is up to their government, but it seems imperative that they do so in a way which will protect their citizens’ health whilst minimizing economic and social disruption, and respecting human rights [8]. The following case study aims to showcase Australia’s initial (January to April) situation and response to COVID-19.

## 2. Case Presentation

### 2.1. Demography and the Australian Context

Australia is a large island nation situated between the Indian Ocean and the South Pacific Ocean. Historically, human occupation began approximately 65,000 years ago, arriving from the Indonesian archipelago and Papua New Guinea. Prior to the arrival of modern Europeans, Indigenous Australians had established populations across the continent [10].

Approximately 70% of the continent is arid (Australia is the driest continent in the world apart from Antarctica), with 35% of the landmass classified as desert [11]. The remainder of the continent ranges from tropical and temperate forests through the Northern Territory and down the eastern seaboard, to snowy areas in the Australian Alps [12].

From the arrival of the first fleet in 1788, six British colonies were established across the country which united as states to form the Commonwealth of Australia in 1901. These states retained their original systems of government and agreed to defer to the federal government responsible for matters concerning the commonwealth as a whole [13].

With an area more than 50% larger than Europe, and a current population of 25.6 million people, Australia has one of the lowest population densities in the world at 3.3 people per square kilometre. Nearly 90% of the country live in urban areas primarily concentrated along the eastern seaboard, with two thirds of the population living in one of the state capital cities [14].

Australia is a multicultural society, with 33% of the population born overseas. A total of 18.7% of the population are below 15 years of age, 65.6% are aged 15–64, and 15.7% are over 65. The average Australian household contains 2.6 people. Diversity is highlighted by the fact that 21% of the population speaks a language other than English at home, and 3.3% of the population identify as Aboriginal and Torres Strait Islander [15].

Australia has a large, highly developed market-based economy with a long history of sustained stable growth. With a current GDP per capita of USD 51,701, Australia is among the wealthiest countries in the world. Australia has substantial natural resources and operates at a trade surplus. Top exports include iron ore, coal, natural gas, education (foreign students) and short-term tourism. Prior to March 2020, the key challenges to the Australian economy related to its lack of progress in addressing climate change and biodiversity loss, elevated housing prices, and inequalities and increasing poverty related to the indigenous population [16], who experience significantly worse health outcomes than the general population [17].

Australian society is egalitarian, highly individualistic with low power distance, competitive and meritocratic, with an intermediate tendency to avoid uncertainty [18]. These cultural tendencies will impact on the effectiveness of implementation of non-pharmaceutical interventions that are implemented by Australian authorities.

### 2.2. Health Care System

Australians are living longer and with better health than ever before. Most report they are in good to excellent health, and overall Australians have close to the highest life expectancy in the world (for both newborns and at age 65). Prior to February 2020, the policy focus was on addressing non-communicable diseases (NCD), particularly cancer, respiratory diseases like asthma, Type 2 diabetes, and heart, stroke, and vascular disease [17].

Over the past 10 years, Australian’s health status has generally improved in terms of infant mortality and life expectancy, and incidence of heart attack, bowel cancer, and Hepatitis A and B infection. Over the same period, there has been an increase in incidence of syphilis, chlamydia, and gonorrhoea, and an increase in hospitalisation for injury and poisoning. The determinants of health in Australia have also shown improvements in healthy behaviours (particularly a reduction in the proportion of adults who are daily smokers), improvements in educational attainment, and a reduction in the proportion of people with low income. Unfortunately, there have also been some unfavourable changes in health behaviours, most significantly an increase in the proportion of the population who are overweight and obese [19].

In 2017–2018 Australia spent a total of AUD 185.4 billion on health care, or just about 9.3% of GDP. This is slightly above the OECD average—equivalent to New Zealand, Brazil, Portugal and Finland, and well behind Switzerland (12.2%) and the United States (16.9%) [20]. The Australian health system funding is complex, with services funded by the Commonwealth Government, state governments, private health insurers (both for and not for profit), and individuals via out-of-pocket expenses.

Medicare is Australia’s universal health insurance scheme. Managed by the Commonwealth Government it covers all citizens, permanent residents, and their families. Medicare primarily funds access to medical practitioners (through the Medical Benefits Schedule), medicines (through the Pharmaceutical Benefits Scheme) and some diagnostic imaging [21]. Under Medicare all Australians are eligible to be treated in a public hospital free of charge. Most visits to General Practice (Primary Care or Family) physicians incur no out-of-pocket charge as the costs are fully covered by the MBS; however, GPs are allowed to set their own fees and in some instances there is a gap above the MBS rate that individuals will need to personally cover.

State Governments are responsible for accrediting, running, and funding public hospital and community health services, although they do receive commonwealth contributions to these services. States pay for hospital services via activity-based funding, where they set an efficient price for each activity of care based on the procedure and the circumstances of the patient [22]. Due to shared responsibilities and overlapping accountabilities, health service organisation and funding is highly political in Australia, with the state and commonwealth governments frequently “blame shifting” and accusing each other of being responsible for resource shortfalls [17]. This impedes coordination and collaboration across the whole system.

Recently, a new ranking system designed to rate health system performance has been proposed [23]. This ranking evaluates performance using nine indicators across three key domains—general performance, major clinical performance, and health system equity and sustainability. In this system, each indicator is scored from 1 to 3, aggregated to a score out of nine for each of the domains.

Each of the domains is graded A if the resulting score is 8 or 9, and by this methodology we can see that Australia’s health system is rated AAA (Table 1). This compares favourably with other health systems rated to date, including Switzerland, Germany, France, and Japan which all are graded as AAB.

Overall, Australians enjoy a generally good quality of health and experience a reasonably effective health service. The key system challenge relates to coordination of activities and services across a complex organisational and funding domain. Political constraints have previously impeded better coordination of services from the patient perspective, and while individual organisations are starting to evaluate their services from a patient centred perspective there is still a long way to go. Addressing the COVID-19 pandemic will require a degree of responsiveness and flexibility which to date has not been demonstrated by the Australian health care system.

## 3. Epidemiology and Mathematical Modelling Prediction

As of 20 April, Australia had not yet reached “widespread community transmission of COVID-19” [9]. Thus far, 431,000 tests had been performed, of which, 1.5% of tests were positive. For comparative purposes, the USA had a test positivity rate of almost 20% in April [30]. Additionally, Australia’s total tests conducted versus confirmed cases compared favourably with countries that demonstrated commendable national response strategies such as New Zealand and South Korea. As of 20 April 2020, Australia had 6619 confirmed cases, 72 deaths, and 4258 recoveries. Of the confirmed cases, most infections (64%) were acquired internationally due to recent overseas travel. Thirty-two percent of cases were acquired in-country. The majority of local transmission was via direct, close contact with infected individuals and concurrent public health investigations were conducted to determine additional sources of infection [9].

The first four cases of COVID-19 in Australia were reported on 25 January 2020 in New South Wales (3) and Victoria (1). Australia reached its 100th case on 10 March 2020 and numbers rose exponentially [9] (Figure 1). In the early stages of the pandemic outbreak, COVID-19 had an estimated basic reproduction number (R0) between 2 and 3 [31]. By the end of March, the number of reported COVID-19 cases doubled every 3.4 days and had a cumulative incidence growth rate above 20% per day (Table 2) [32]. Had the doubling time continued along that trajectory, the number of reported cases in Australia would have reached 100,000 before mid-April (Figure 2). The mitigation measures implemented by the Australian government and the compliance of the general public have assisted in slowing the spread of the virus and the flattening of the curve (Figure 2). The first wave peak of reported cases occurred on the 29 March, with a total of 527 daily cases. Since 30 March, the number of cases and doubling time in days consistently decreased (Table 2). As on the 20 April 2020, the doubling time of COVID-19 within Australia was 112 days. New South Wales, South Australia, and the Australian Capital Territory had lower doubling times than the national average. Tasmania’s doubling rate was much lower than the national average and the other states.

Up until 29 March 2020, Australia’s exponential growth rate followed that of the United Kingdom’s (Figure 3). Due to the government’s early lockdown and implementation of non-pharmaceutical measures (see Section 4.1), Australia managed to slow the growth rate and “flatten the curve”. Britain instigated mitigation interventions later than Australia and, consequently, their reported number of cases was not far behind Italy’s, which had an initial doubling time of two days. While Singapore and Japan are known for strict case isolations and contact tracings, their daily number of cases were nonetheless gradually rising—possibly due to more lenient social distancing measures. Figure 3 provides a comparison between Australia and other countries’ national trajectories. The relatively lower rates of community transmission within Australia (32%) during the initial months of the pandemic helped prevent Australia’s healthcare system from becoming overwhelmed [9].

An ACEMod model demonstrated that the closure of schools in Australia would have limited effects on containing and reducing the peak of the virus in the early stages of the pandemic in Australia [32]. The Australian Census-based Epidemic Model (ACEMod) is a stochastic agent-based and discrete-time model that simulates potential outbreaks scenarios in Australia. It was developed for the simulation of influenza outbreaks. ACEMod is adjusted to the 2016 Australian Census Data [33]. School closures would produce a two-week delay in the epidemic’s peak, allowing healthcare facilities to better prepare accordingly. This model also predicted that social distancing would suppress the spread of COVID-19, but only if compliance levels were 80% or higher and were to last 13 weeks. International travel restrictions and case isolations would reduce the incidence peak by 24% and the prevalence peak by 22% (at a predicted epidemic peak seven weeks after the 1000th case); however, these measures would need to be implemented alongside social distancing. It also suggested that recurring seasonal wintertime outbreaks are possible [34]. Viral transmissibility is higher during wintertime in temperate zones of the Northern and Southern Hemispheres [35]. A different model conducted by Neher et al., (2020) predicted a peak of COVID-19 transmissibility during the winter of 2020 in the Southern Hemisphere. Although Australia managed well in slowing the spread of the virus, mitigation measures should continue throughout the upcoming winter months in light of these findings. The mid- to long-term control of the COVID-19 epidemic in Australia may also be affected by—in addition to the mitigation measures discussed above—“the degree of seasonal variation in transmission, the duration of immunity, and the degree of cross-immunity between SARS-CoV-2 and other coronaviruses” [34]. As with any disease outbreaks, to correctly simulate and predict the incidence and transmission of COVID-19 is difficult. Modelling for potential scenarios, with a combination of mitigation and suppression measures, allows for appropriate planning and timely interventions [31].

### 3.1. SEIR Mathematical Modelling

The following predictions made using SEIR mathematical model may give insight to the duration of the COVID-19 pandemic in Australia and the maximum number of infections we could expect if current mitigation measures fail and Australia experiences exponential growth once again. For the implementation of these models, we assumed that the Australian population is homogeneous and constant. The SEIR model divides the given population into four categories; (S) Susceptible—(E) Exposed—(I) Infectious—(R) Recovered/Removed. Figure 4 shows theses four categories and the rate at which these groups move to the next category [33]. Compared to the classical SIR model, the SEIR Model includes the additional incubation period denoted as σ, including the population of individuals who have possibly been infected (E exposed) but are not yet infectious themselves.

The parameters of the SEIR model used in our study are listed in Table 3. We used a rounded figure (N = 25,600,000) for the current population of Australia [37]. The initial number of confirmed COVID-19 cases in Australia was 4. However, the initial number of exposed individuals in Australia was difficult to allocate and therefore we have used the estimated ratio of 2.399 submitted to the WHO [38]. This number indicates that 2.399 individuals are exposed for each infected individual. All calculations and graphs have been generated in Microsoft Excel.

In order to determine ß, the rate at which the infection is spreading, we take the basic reproduction numbers of two recent Australian studies, R0 = 2.27 and R0 = 2.53 [32,39], and plot these predictions against our data of actual confirmed COVID-19 cases as of 17 April (Figure 5). 

We can assume from the current trajectory of actual cases; the basic reproduction number lies closer to 2.27. 

Using the number for R0 we can derive ß using the following equation [38]:R0 = ß/ƴ(1)

ƴ depicts the recovery rate assumed to be 0.147 (Table 3), given that the average total duration of infection is 12 days and incubation period (*Y*) is 5.2 days [38,40,41].

To obtain the incubation rate (*σ*), we use the following formula [38]:*σ* = 1/*Y*(2)
(3)∴ σ=1÷5.2=0.192

### 3.2. SEIR Mathematical Modelling Results

By mid-April Australia had managed to flatten the curve. It must be noted that the following SEIR models are predictions for situations experiencing exponential growth and therefore may not be applicable if Australia continues to flatten the curve. However, had the early mitigation measures not been able to contain the spread of the virus, Australia could indeed have experienced the following predictions. 

As seen in Figure 5, Australia’s trajectory was between R0: 2.27 and R0: 2.53 in mid-April. We consequently created models for both scenarios (Figure 6). At on R0 of 2.27, the number of COVID-19 cases would surge to a maximum of 2.9 million cases (n = 2,914,390) on day 180 of Australia’s outbreak onset, which would occur around mid-July 2020. At a slightly higher R0 of 2.53, 2.9 million cases will be reached almost 20 days earlier on day 168. If Australia’s current trajectory stayed within or below an R0 of 2.27, the expected number of infections was predicted to be 15,399 by the end of April 2020. Had the early mitigation measures not contained Australia’s trajectory, an increased R0 of 2.53 would have led to three times the predicted number of infected people (~51,620 infections). In reality, Australia managed to keep infection rates well below an R0 of 2.27, confirming less than 7000 cases by the end of April 2020.

Although the SEIR model incorporated a latency period, accounting for those individuals exposed to the disease but not infected, the model has its limitations. It must be noted that the assumptions made here are based on a homogeneous population and assumed all individuals are affected equally, it does not represent an authentic, real-life scenario. To improve the model with realistic outcomes, consideration needs to be given to the heterogeneity of the Australian population, climate, and effects of additional mitigation measures such as social distancing.

## 4. Discussion

### 4.1. Non-Pharmaceutical Interventions

Following the recommendations of the Australian Health Protection Principal Committee (AHPPC), intervention measures were implemented nationwide to control the transmission of COVID-19 (Table 4) [42,43,44,45,46,47,48,49,50,51,52,53,54,55,56,57]. Since January 2020, certain travellers were asked to self-isolate themselves, initially at home and then it was changed to designated facilities ideally at the city of arrival, for 14 days upon return from an affected area [42]. This measure, first implemented with travellers from Hubei Province of China (January 29), was subsequently extended to all of mainland China (on 1 February) as well as Iran (1 March) [43,44,45], Republic of Korea (5 March), Italy (11 March), and to all countries (15 March) [46,47,48]. 

In March 2020, the Australian Government progressively enforced travel restrictions and travel bans which led up to a border closure to all non-citizens and non-residents [49]. Cruise ships arriving from abroad were banned from Australian ports [47]. Some states such as Tasmania, the Northern Territory, Western Australia and South Australia have closed their borders and required those who were allowed to enter to self-isolate for 14 days [57].

Several measures were taken to limit the number of people in non-essential outdoor and indoor gatherings as well as restrictions on visitors and staff into aged care facilities [50,51]. In order to slow down the spread of COVID-19, further measures were placed a few days after the initial restrictions on gatherings, including closure of some social gathering venues [52]. With regard to securing the supply of personal protective equipment, all non-urgent elective surgeries were postponed [53]. People were encouraged to stay home unless for the purpose of shopping for essentials, health or travelling to work or education [58].

### 4.2. Observed and Expected Economic Impact

The full impact of COVID-19 on the Australian economy could take months or years to materialise and it is difficult to predict the sheer extent of economic damage. Stock prices steadily declined between January and April and the Australian dollar dropped 13% between early January and mid-March [59].

The impact of COVID-19 on Australia’s economy is predicted to result in two or more quarters of negative growth [60] and may bring Australia’s economy to its first recession in 28 years [61]. The coronavirus outbreak within the country detracted 0.3 per cent from economic growth for the March quarter [62]; the country’s economy had already decreased by 0.2 percentage points due to the summer’s damaging bushfires. The accommodation and food services industry were hit the hardest throughout the March quarter; 78% of businesses reported decreased revenue due to COVID-19 in March and 84% expected a “reduction in demand for goods and services in the next two months” in April [63]. 25.6% of employees had to be let go within this industry between mid-March and early April and jobs reduced by 33.4%. Retail turnover decreased by an unprecedented 17.9% in April; cafes, restaurants, and takeaway food services were the most impacted. According to the ABS’s survey on *Business Impacts of COVID-19*, approximately half of Australian businesses reported having to reduce staff working hours, ask employees to work from home and place individuals on leave [64]. A KPMG Economics model estimates that a total of 2 million Australian workers are economically affected by the virus. Factoring in disease severity and recovery time, a 1.22 per cent loss in economic productivity is predicted [60]. 

On 22 March 2020, the Australian federal government released its second fiscal package (66.1 billion AUD support package) in response to the coronavirus pandemic. In combination with the Reserve Bank of Australia’s AUD 90 billion loan fund (at 0.25% interest), previous fiscal (17.6 billion) and healthcare packages (2.4 billion), and states and territories own economic stimuli, the total economic support amounts to 189 billion AUD, 10% of the country’s GDP. The stimuli packages are targeted at welfare recipients, workers, households, and businesses and will ensure the flow of credit within the Australian economy. Funding to higher education providers and aid for affected workers to upskill and retrain is provided by a Higher Education Relief Package [57]. The Commonwealth Government says the support packages will support the economy by “maintaining confidence, supporting investment and keeping people in jobs” [59]. In order to facilitate job searches for the recently unemployed, the government launched a website (Jobs Hub) that advertises current vacancies within Australian businesses. KPMG models predict it may take up to ten years for the Australian GDP to return to the forecasted pre-COVID-19 levels [60]. 

Lastly, lower tourism (the tourism industry contributes 3 per cent of Australia’s GDP) and international student expenditure will continue to contribute to poor growth in the upcoming months [59]. Revenue associated with tourism and higher education fees are expected to drop by 25% and 15% respectively for 2020. Overseas arrivals to Australia in April dropped by 99% when compared to April 2019 [63]. 

Considerable uncertainty remains concerning the effects of COVID-19 past the June 2020 economic quarter and the full extent of the economic impacts are unknown. The economic shock, nevertheless, as seen globally, is likely to be significant. Government support should be directed to the health sector and aim its complementary policy actions to minimise economic and social fallout.

### 4.3. Media Coverage 

Media coverage of COVID-19 was abundant in the weeks following Australia’s first case of COVID-19 on 25 January 2020. Australia’s official government website [65] featured the latest COVID-19 news, updates and advice from government agencies across Australia along with press briefings and transcripts of addresses made by Australia’s current Prime Minister, Scott Morrison. Furthermore, national and international news of COVID-19 along with information from leading health professionals (such as the WHO), ministries of health, and government advice dominated online platforms such as News.com.au and the BBC (Australia). Further still, free-to-air broadcasters such as Channel 7,9 and TEN dedicated special programmes with health experts sharing the latest information on COVID-19 and Australia’s ABC News live streamed a Q&A session for the general public to ask health experts questions relating to the virus. However, irresponsible media coverage of the COVID-19 also circulated and contributed to unnecessary fear and panic within the Australian public [66].

### 4.4. Social and Political Disruption

Social disruption has been severe and widespread since early March 2020. At this time, more countries were imposing stricter quarantine measures such as border closures, banning of sporting events and closures of schools and universities in order to contain the spread of the virus. These significant social disruptions seen globally were a predictable disruption to the Australian lifestyle. By mid-March when Australia placed travel bans on Iran, Korea and Italy, rumours spread across social media of a nationwide lockdown [67]. Australians were seen emptying supermarket shelves of pasta, rice, and toilet paper and the popular Twitter hashtags #toiletpapergate, #coronapocalypse circulated social media platforms. This high presence on social media allowed for the rapid spread of inaccurate information and dissemination of rumours, a phenomenon the WHO referred to as an “infodemic” [68]. By mid-April 2020, Australian employee jobs decreased by 7.5% [69]. This high traffic of newly unemployed Australians caused the welfare claims website to crash forcing Australians to form large queues outside Centrelink offices in order to claim their unemployment benefits. These lines did not respect social distancing measures and images soon circulated social media platforms such as Twitter with hashtags #COV19au and #centrelink. As the COVID-19 pandemic unfolds in Australia, how the general public acclimate to mitigation measures and how the Australian Government addresses the general public is crucial in diffusing panic [70].

Australia’s strategy to minimise the expected degree of disruption involved communication at a National level and coordination with state and territory governments and health sectors to respond promptly and effectively to manage an influenza pandemic [71]. Additionally, the Australian Government published their first Australian Health Sector Emergency Response Plan for Novel Coronavirus (the COVID-19 Plan) on 18 February 2020 [72]. Aligned with the WHO’s International Health Regulations (IHR) [73], the plan outlines a comprehensive risk communications strategy designed to reach a broad range of stakeholders including health authorities, the medical profession, media and the public. Risk communication via media channels is not only an essential intervention to manage infodemics but also a key pillar in a country’s response to an outbreak in order to prevent international spread of disease and preserve social, economic and political stability [68]. 

Such communication provides real-time information exchange and advice between health experts, government officials and the public with the aim of informing people on how to mitigate the effect of a disease outbreak and take protective and preventive action. Through open, accurate and transparent risk communication—which is consistent, clear and timely—Australians will inevitably adapt to this new and most likely temporary lifestyle [68].

### 4.5. Possible Exit Strategies

Since the start of the year, unprecedented steps have been taken globally to combat the spread of the COVID-19 disease. In Australia, as previously described, significant non-pharmaceutical interventions have been implemented to minimise transmission and reduce the burden on the health system arising from the pandemic. These interventions include closures of borders, non-essential businesses and gatherings, schools, places of worship, and strict social distancing measures [58].

These interventions appear to have worked, with reductions in daily incidence of COVID-19 disease in every state and territory. In parallel, health services are being reinforced, with expanded intensive care capacity and increases in ventilator numbers [74], and greater access to personal protective equipment [75]. 

Ultimately pandemic conditions are likely to persist until either a level of herd immunity is achieved, there is a cure or an effective way of medically managing cases, or until either there is an available and effective vaccine [76]. The current question regards Australia’s next steps and how the exit from the restrictions can be managed.

In May Australia had a window of opportunity to pursue an elimination strategy. This is a course of action where the goal is to implement strict measures to reduce the R_0_ < 1 for long enough that transmission of the disease is eliminated in the population. From this point forward strong testing, contact tracing, and isolation measures need to be in place to rapidly identify new imported cases of the disease, to isolate the infected, and to quarantine the exposed [77]. This strategy requires strong border controls, capacity for rapid industrial scale testing, adequate supportive capacity in the health system, and robust processes for managing quarantine and isolation. Above all, the Australian people must be willing to continue to surrender their personal liberties as required to prevent disease transmission. 

If implemented robustly, this process should ensure that once current non-pharmaceutical measures begin to be relaxed there will not be further waves of infection and society and the economy can begin to restart. There is some evidence from China that this could be the “least bad” option [41]; however, a key challenge is that no herd immunity will be established leaving Australia exposed to the potential of future epidemics unless constant vigilance and preparedness are maintained. This is the option that New Zealand is currently actively pursuing [78].

The nearest alternative to the elimination option is one of suppression. This involves taking enough steps to “flatten the curve” or reduce the daily incidence to the level required to ensure that the health system is not overwhelmed. This approach accepts that a level of community transmission may occur and seeks to implement targeted measures to keep this transmission to an acceptable rate that ensures that health services do not exhaust available capacity [79]. Depending on the level of community transmission and the developing capability of the other measures in place, this approach does not necessarily exclude the possibility of achieving elimination over time.

Politically, this is a challenging path to take as it means taking decisions to loosen current restrictions that will actively harm some people. While the economic impact is currently unprecedented, what is less clear is the longer-term effect of this economic impact on population health. As of 21 April 2020, policy makers appeared to be happy to trade the known harm of relaxing restrictions with the unknown harm of the longer-term economic impact.

The Australian Commonwealth and State Governments are pursuing what they call a suppression/elimination exit strategy [80]. The current goals of the response to this pandemic are to minimise the infection rate, minimise mortality, and manage demand on the health system. Given these goals and the high levels of economic support that have been outlined, it was feasible for Australia to follow the New Zealand approach of working towards elimination at least in the short to medium term; however, acknowledging the complexities of the elimination option, the government is taking a flexible approach. The feasibility of an elimination in Australia, a much larger country than New Zealand, remains to be seen.

## 5. Conclusions

The observed impacts of the global pandemic are still emerging, due to multiple major unresolved clinical and public health issues such as the sustainability and feasibility of interventions [81]. Once this crisis is over, further studies can be conducted on its aftermath on Australia, including the social, political, and economic aspects. Australia has also followed the lead of many other countries to implement travel restrictions against the WHO’s advice. Despite the IHR being legally binding, many countries still violate the guidelines when imposing travel restrictions [82].

According to an epidemic modelling assessment based on real-world data (Global Epidemic and Mobility Model), the strict travel restrictions on Wuhan only modestly slowed down the spread of the virus in Mainland China by less than a week. However, a more significant effect of a few weeks’ delay was observed internationally. This model also showed that travel restrictions are likely to be less effective to contain the disease unless used in conjunction with interventions that reduce COVID-19′s transmissibility [83].

A similar model on Australia’s travel bans and transmissibility reduction could be conducted to obtain potential data to guide policy making decisions regarding public health measures to contain COVID-19 in the future. Our analysis of the cumulative reported COVID-19 cases showed that the curve in Australia had flattened by the beginning of April 2020.

The Australian community continues to live with constant disruption to their daily lives as they adapt to the changes in interventions as well as their fear of contracting the virus. With several COVID-19 vaccines currently in early developmental phases, the best practice right now is implementing non-pharmaceutical measures—including, but not limited to, hand washing, cough etiquette and social distancing—until more information is available on this infectious disease. The ultimate aim of these preventative measures is to reduce the transmission of the virus, to flatten the epidemic curve, and collectively greatly reduce the risk of overburdening the health care system in Australia. In addition to traditional media, social media is also a platform to deliver public health information in today’s digital word. It is important to be updated with the latest information as the situation is developing. There is an urgent call for a transdisciplinary collaboration to fight and end this pandemic.

Following the modelling presented in this case study, some areas of Australia have experienced a second wave through the cooler winter months from May through to October 2020, particularly in the southern state of Victoria [84]. Further investigations should be undertaken to examine the effectiveness of more stringent non-pharmaceutical interventions, including the mandatory wearing of masks and significant restrictions on the movement of people on the local course of the disease.

## Figures and Tables

**Figure 1 ijerph-17-08939-f001:**
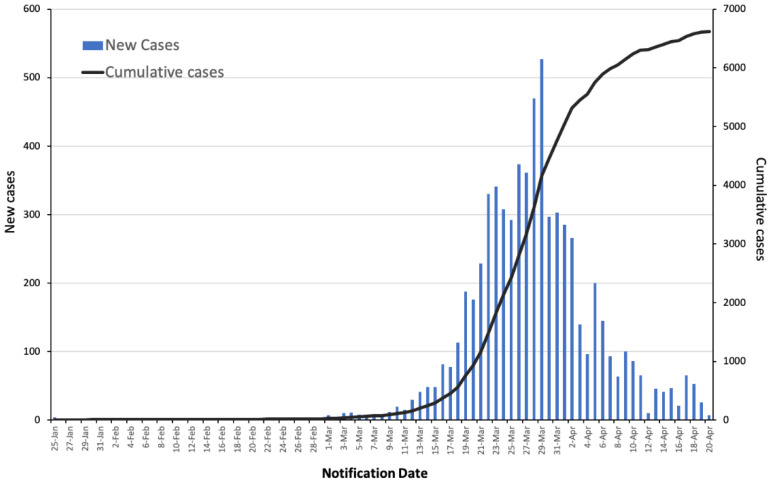
New and cumulative reported COVID-19 cases. Data retrieved from the Australian Government Department of Health (2020). Last updated on 20 April 2020 [9].

**Figure 2 ijerph-17-08939-f002:**
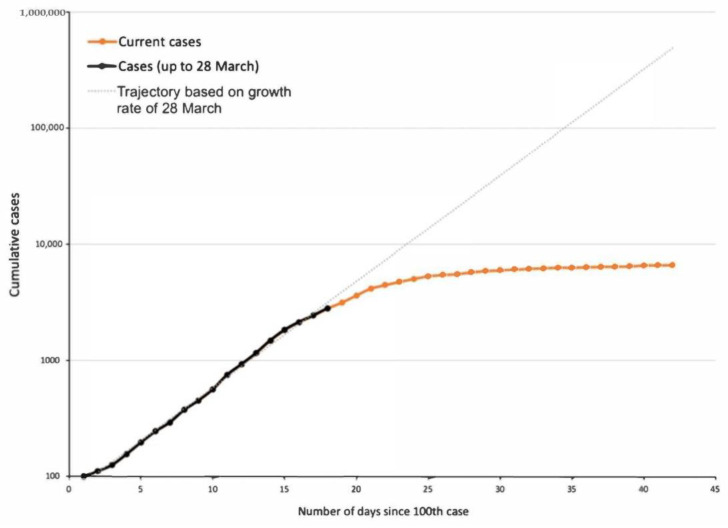
Cumulative growth of COVID-19 cases in Australia (shown since 100th case). The dotted line represents the possible trajectory of COVID-19 cases based on the growth rate of 28 March 2020. Last updated on 20 April 2020 [9].

**Figure 3 ijerph-17-08939-f003:**
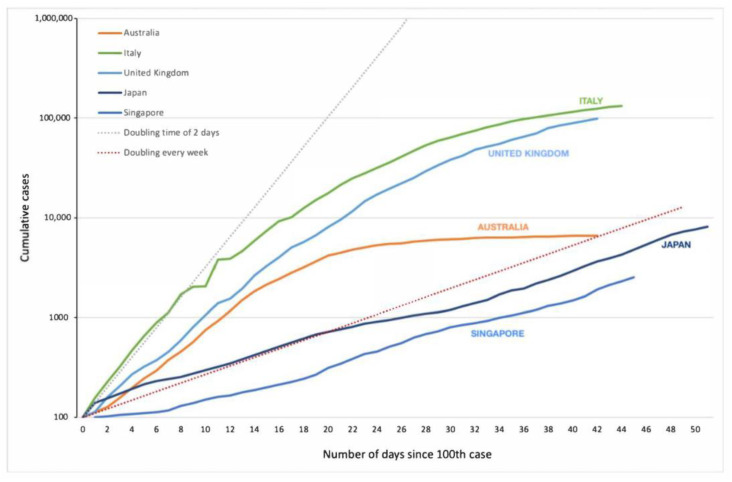
Logarithmic scale of cumulative reported cases of COVID-19 in Australia, Italy, the United Kingdom, Japan, and Singapore. Dotted lines represent different COVID-19 doubling rates. Country data retrieved from the John Hopkins Coronavirus Resource Centre [36]. Last updated on 20 April 2020.

**Figure 4 ijerph-17-08939-f004:**
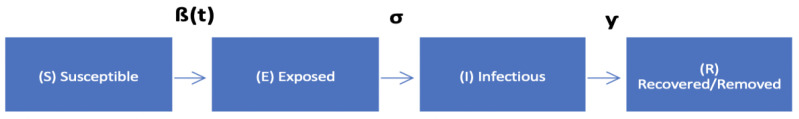
SEIR Model with rate of movement between categories.

**Figure 5 ijerph-17-08939-f005:**
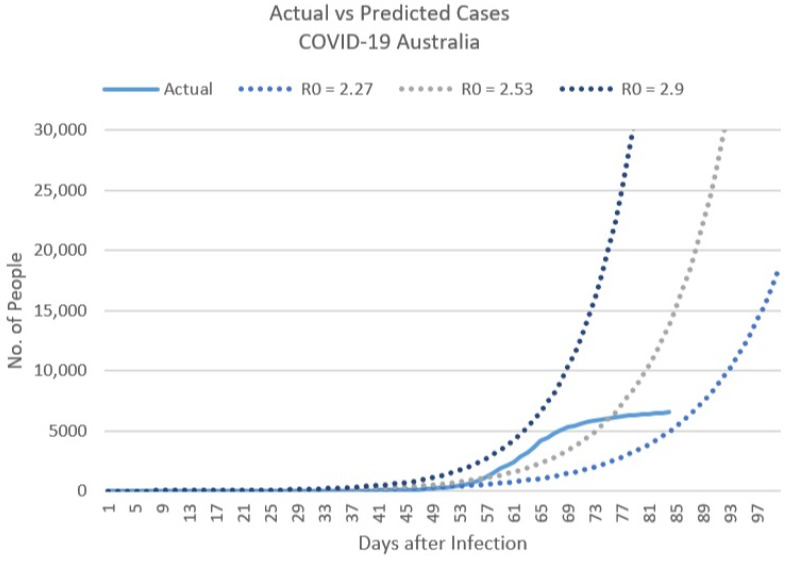
Actual vs. Predicted COVID-19 confirmed cases as of 17 April 2020.

**Figure 6 ijerph-17-08939-f006:**
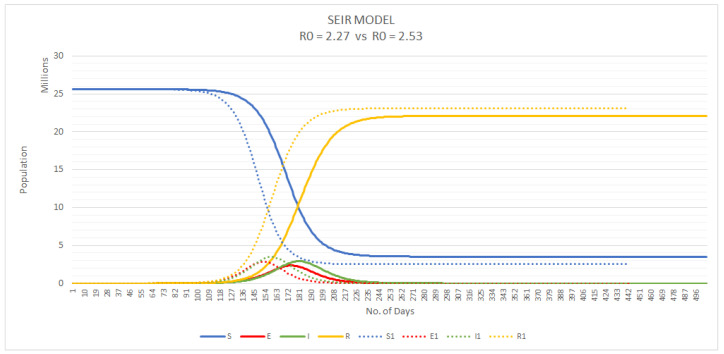
SEIR model predictions with basic reproductive number (R0) of 2.27 (solid lines) and 2.53 (dotted lines).

**Table 1 ijerph-17-08939-t001:** Ranking the Australian Health System.

	Performance Indicators	Target Performance (Score = 3)	Towards Target Performance (Score = 2)	Far Below Target Performance (Score = 1)	Australia Score	Rating
Domain I	UHC ^1^—Beyond Secondary Health Care Services provided to (% of the population)	>90%	50–90%	<50%	>90% * [24]	3
	Female’s HALE ^2^ at Birth (in years)	>70	50–70	<50	74.4 years [25]	3
	Number Surgical Procedures Performed (% of the Minimum Number Needed)	>90%	50–90%	<50%	184% [26]	3
	General Performance Score					9
Domain II	Overall HAQ ^3^ Index	>90	50–90	<50	95.5 [27]	3
	HAQ index for Hodgkin’s Lymphoma	>90	50–90	<50	100 [27]	3
	% 30 days-survival of Patients Hospitalized with Acute MI ^4^	>95%	85–95%	<85%	96.20% [20]	3
	Clinical Performance Score					9
Domain III	% of Accredited Hospitals Providing Tertiary Care	>90%	50–90%	<50%	>90% ** [28]	3
	Financial Protection, Health Care Equity and Funding Sustainability (% of the Population)	out of pocket expenditure on health care less than < 5% of total household income	out of pocket expenditure on health care 5–7.5% of total household income	out of pocket expenditure on health care >7.5% of total household income	5.13% [29]	2
	Gov. Funding for Health Research (% of GDP ^5^)	>0.2%	0.05–0.2%	<0.05%	0.366% [21]	3
	Health System Equity and Sustainability Score					8

^1^ UHC = Universal Health Coverage, ^2^ HALE = Healthy Life Expectancy, ^3^ HAQ = Health Access and Quality, ^4^ MI = Myocardial infarction, ^5^ GDP = Gross Domestic Product, * All Australians are covered by Australia’s universal health insurance scheme—Medicare, ** All public and private hospitals, day procedure services and public dental practices are required to be accredited to the National Safety and Quality Health Service Standards.

**Table 2 ijerph-17-08939-t002:** Doubling rate of COVID-19 in Australia and by states and territories. Doubling rates calculated from the growth rate of the last 10 days. Data retrieved from the Australian Government Department of Health (2020) [9]. Last updated on 20 April 2020.

Doubling Time (Days)	Australia	NSW	VIC	QLD	SA	WA	TAS	ACT	NT
28 March	3.40	3.47	4.01	3.88	3.10	3.37	3.34	2.04	NA *
8 April	21.26	17.6	17.5	17.1	22.6	20.0	14.3	25.6	NA *
17 April	85.6	92.4	61.3	75.3	157.5	52.9	10.6	138.6	NA *
20 April	111.8	141.4	110.0	105.0	385.1	91.2	15.8	1732.9	NA *

* Not available. Northern Territory has had no new cases for the past two weeks (28 overall cases—20 April 2020).

**Table 3 ijerph-17-08939-t003:** SEIR parameters required to predict the COVID-19 pandemic for Australia. Inputting these variables into the equations will give us the parameters needed to make our predictions using the SEIR model.

Equations	Values	Definitions	R0 = 2.27	R0 = 2.53
R0 = ß/ƴ		S = Susceptible; initial	25,599,986	25,599,986
*β* : rate of spread; β=R0×γ		E = Exposed; initial	9596	9596
*Y*: duration of incubation	5.2 days	I = Infected; initial	4	4
σ: rate of latent (exposed) individuals becoming infectious σ = 1÷Y;	0.192	R = Recovered; initial	0	0
D: average duration of recovery D= total duration−incubation period (Y)	6.8	ß = Rate of spread of infection	0.334	0.372
γ: Recovery rate γ=1÷D	0.147	σ = Incubation rate	0.192	0.192
$Sn=Sn-1-\left(\left(Sn-1_{}÷S\right)\times \left(ß\times In-1\right)\right)$		ƴ = Recovery rate	0.147	0.147
$En=En-1+\left(Sn-1_{}÷S\right)\times \left(ß\times In-1\right)-\left(En-1_{}\times \sigma \right)$		T = time interval; usually days	1	1
$In=In-1_{}+\left(En-1_{}\times \sigma \right)-\left(In-1_{}\times ƴ\right)$		n = number of people on day n	varies	varies
$Rn=Rn-1+\left(In-1_{}\times ƴ\right)$		N = total number of people = S + E + I + R	25,600,000	25,600,000

**Table 4 ijerph-17-08939-t004:** National non-pharmaceutical intervention measures implemented in Australia during the COVID-19 pandemic.

*Date of Implementation*	*Measure(s)*
*29 January*	14-day self-isolation for travellers arriving from Hubei Province of China
*1 February*	Travel ban on mainland China *; 14-day self-isolation for travellers arriving from mainland China
*1 March*	Travel ban on Iran *; 14-day self-isolation for travellers arriving from Iran
*5 March*	Travel ban on Republic of Korea *; 14-day self-isolation for travellers arriving from Republic of Korea
*11 March*	Travel ban on Italy *; 14-day self-isolation for travellers arriving from Italy
*15 March*	Universal precautionary 14-day self-isolation requirement on all international arrivals Arrival ban on cruise ships from foreign ports for 30 days **
*16 March*	Limit of fewer than 500 people for non-essential, organized public gatherings
*18 March*	Limit of fewer than 100 people for non-essential indoor gatherings Limit of fewer than 500 people for outdoor gatherings Restrictions on visitors and staff into aged care facilities Overseas travel restrictions for Australians
*20 March*	Border closure to all non-citizens and non-residents *; 14-day self-isolation for all travellers
*23 March*	Closure of social gathering venues including pubs, clubs, hotels, gyms, indoor sporting venues, hotels, cinemas, entertainment venues, casinos, night clubs and places of worship Restrictions on funerals (adhere to 1 person per square meter rule) and restaurants and cafes (takeaway and/or home delivery only).
*25 March*	Enhancement on social distancing measures by prohibition of additional activities and venues Postponement of all non-urgent elective surgery in public health systems
*28 March*	14-day self-isolation for all travellers at specified facilities
*1 April*	Postponement of all non-urgent elective surgery in private hospitals

* with exemption of Australian citizens, permanent residents or others exempt from entry restrictions, ** provisions made for cruise ships already en route to Australia such as a 14-day self-isolation on Rottnest Island

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
