# Peer review of "COVID-19 Down Under: Australia’s Initial Pandemic Experience"

_ijerph, 2020, doi:10.3390/ijerph17238939_

Round 1
Reviewer 1 Report
“COVID-19 Down under: Australia’s initial pandemic experience” is a very well written and presented manuscript describing, as the title indicates, Australia’s experience in the early stages of the pandemic caused by SARS-Cov-2. The authors have provided a very thorough yet succinct overview of Australia’s demography and health care system, which really adds to the value of this manuscript and provides appropriate context. The manuscript also presents the findings of a SEIR modelling conducted using data from the first wave of the epidemic.
Although overall, I support the publication of this manuscript I do have one major criticism or concern that needs to be addressed prior to publication. Due to the highly evolving nature of this pandemic much of the information presented in the manuscript is outdated. I realise this is a case report, and although the authors do make some mention of the changing situation in the concluding paragraph, I still feel that the manuscript needs an overhaul to incorporate this changing situation and experiences to-date. I am not suggesting that the authors rerun the model but rather discuss the results in a past tense and update the manuscript taking into consideration the situation and experience since April and how it has changed and then address how these changes might impact the outcomes reported for the SEIR model. This would make for a very interesting and more relevant manuscript that would add scientific value to the COVID-19 pandemic country experience.
Some additional minor comments/edits suggested below:
- I think it would be better to use past tense throughout the manuscript.
Introduction
- Mention that COVID-19 is caused by severe acute respiratory syndrome coronavirus 2 (SARS-Cov-2)
- Novel coronavirus disease 2019 (COVID-19)
Line 56 – apostrophe is missing (government’s)
Line 179 – 182, Table 2: The value for 8th March in the QLD column is missing a value after the decimal point.
Lines 185-186: I think there needs to be some mention of the lockdown and other measures implemented in Australia. It would also be interesting to mention how these compared to those of other countries. ----I see that these have been outlined later in Table 5 but I think these need to be noted earlier in the manuscript (or least reference to the table).
Lines 289-296: I suggest including the dates of when these travel quarantine restrictions were initiated, at least for national borders.
Line 380: Sentence is missing ‘have’ (“may led…”)
Reviewer 2 Report
This is an interesting article considering current global situation. The scientific rationale is clearly reported, and the article was conceived correctly. However, there are some comments to be address.
- Introduction section is too long: some parts are redundant and out of context. I suggest removing paragraph 2.1; to reformulate and shorten paragraph 2.2.
- The authors should justify the period selected.
- Methods are not clearly mentioned; which is the softwer used? More details on statistical analysis should be added in a separated paragraph;
- table showing the key parameters involved in the model, the initial values of the susceptible population, the exposed population, the symptomatic infected population, and the recovered population, should be added;
- how the authors calculated the transmissibility of latent infection?
- since the strength of this type of model, both for fitting and predicting future dynamics, depends on accuracy in reporting cases, authors should analyze uncertainties in the regulation and assess the degree of variation in parameter estimates;
- the authors should test the model on a simulation group.
- Recent references should be included to strengthen the argument.
- The discussion is too long.
The manuscript can be published in after minor revisions.
Reviewer 3 Report
Authors state that the manuscript scope is “to provide a broad overview of the initial Australian epidemiological situation of the coronavirus disease (COVID-19) pandemic” and, at the same time, to “investigate a scenario assuming infections increase once mitigation measures are lifted” using SEIR mathematical modelling. However, the time frame of the manuscript is the early stage of the pandemic, depicting the situation of March and April 2020. Today, more than six months after, and with a lot of European countries that are experiencing the second wave of the pandemic, such time frame for a case study and for a mathematical model that attempts to predict the future is no longer relevant.
Thus, Authors should update the time frame, maybe focusing on mathematical models to simulate the first wave and the second wave.
Moreover, in line 378-9 Authors write: "it is apparent from countries such as Iran and Italy, that a lax response by the 378 country’s healthcare system led to a loss of containment in the spread of the virus": this is a judgmental statement about Iranian and Italian national response that should be avoided in a scientific text.
Round 2
Reviewer 1 Report
I thank the authors for pointing out that the paper was prepared as part of a series and I feel they have therefore justified their methods and approach. I feel that there remains the issue of relevance but the manuscript still contains information that is useful which deserves to be published. The authors have addressed my other comment regarding the use of past tense and revised the manuscript accordingly. If the Editor feels that the manuscript meets the expected criteria for the special edition then I support the publication of this manuscript.
Reviewer 3 Report
I thank the authors for reminding that manuscript was written for the IJERPH special issue COVID-19 Public Policies Around the World: Lessons Learnt from a Series of Case Studies of the First Pandemic Wave.
I still don't get the point to present a simulation analysis if we already know how the first wave continued in April and May. However, I will accept the paper and leave the decision to the Editor.